# Aerosol-Mediated Spread of Antibiotic Resistance Genes: Biomonitoring Indoor and Outdoor Environments

**DOI:** 10.3390/ijerph21080983

**Published:** 2024-07-27

**Authors:** Nazima Habibi, Saif Uddin, Montaha Behbehani, Abu Salim Mustafa, Wadha Al-Fouzan, Hanan A. Al-Sarawi, Hussain Safar, Fatemah Alatar, Rima M. Z. Al Sawan

**Affiliations:** 1Environment and Life Science Research Centre, Kuwait Institute for Scientific Research, Shuwaikh 13109, Kuwait; 2Department of Microbiology, Faculty of Medicine, Kuwait University, Jabriya 13060, Kuwait; 3Environment Public Authority, Shuwaikh 13001, Kuwait; 4OMICS-RU, Health Science Centre, Kuwait University, Jabriya 13060, Kuwait; 5Serology and Molecular Microbiology Reference Laboratory, Mubarak Al-Kabeer Hospital, Ministry of Health, Kuwait City 13110, Kuwait; 6Neonatology Department, Farwaniya Hospital, Ministry of Health, Sabah Al-Nasser 92426, Kuwait

**Keywords:** aerosols, antibiotic resistance genes, horizontal gene transfer, inhalable air fraction

## Abstract

Antimicrobial resistance (AMR) has emerged as a conspicuous global public health threat. The World Health Organization (WHO) has launched the “One-Health” approach, which encourages the assessment of antibiotic resistance genes (ARGs) within an environment to constrain and alleviate the development of AMR. The prolonged use and overuse of antibiotics in treating human and veterinary illnesses, and the inability of wastewater treatment plants to remove them have resulted in elevated concentrations of these metabolites in the surroundings. Microbes residing within these settings acquire resistance under selective pressure and circulate between the air–land interface. Initial evidence on the indoor environments of wastewater treatment plants, hospitals, and livestock-rearing facilities as channels of AMR has been documented. Long- and short-range transport in a downwind direction disseminate aerosols within urban communities. Inhalation of such aerosols poses a considerable occupational and public health risk. The horizontal gene transfer (HGT) is another plausible route of AMR spread. The characterization of ARGs in the atmosphere therefore calls for cutting-edge research. In the present review, we provide a succinct summary of the studies that demonstrated aerosols as a media of AMR transport in the atmosphere, strengthening the need to biomonitor these pernicious pollutants. This review will be a useful resource for environmental researchers, healthcare practitioners, and policymakers to issue related health advisories.

## 1. Introduction

There has been an increasing impetus to understand the development and spread of antimicrobial resistance (AMR) due to prolonged use and overuse of antibiotics, specifically the broad-spectrum ones, which are crucial medications for treating bacterial infections. The concerns emanate because AMR is persistent and omnipresent, resulting in rapid spread to dangerous levels in several countries [1,2]. The World Health Organization (WHO) has postulated that by 2050, AMR will result in 10 million additional deaths [3]. Thompson [4] reported 4.95 million AMR-related mortalities during 2019, way higher than the deaths claimed by HIV or malaria. The One Health and Global Health concept was proposed by the WHO, International Monetary Fund (IMF), and World Bank (WB) to address AMR-linked problems [5,6]. One Health is a pragmatic and holistic approach considering microorganisms, AMR vectors, and spread and effect at the ecosystem level [6]. The presence of antibiotics in the local aquatic environment [7,8] and antibiotic resistance genes (ARGs) [9] has been associated with the overuse of antibiotics in treating human illness, as well as in the veterinary, agriculture, and aquaculture sectors, and inefficiency of the wastewater treatment plants in their removal.

The presence of ARGs in soil [10,11,12,13], sediments [14,15,16,17,18,19,20,21,22,23], and water [11,24,25,26,27,28,29] has been widely reported. However, air as a medium for ARG dissemination is less explored [30] even though aerosols are known to be carriers of many pathogenic microbes [31,32,33,34,35,36,37,38,39,40,41]. The transfer of microbes and contaminants via inhalation has been reported earlier [33,34,35,40,42,43], so we hypothesize that ARGs too can be transmitted among living beings via bioaerosols. Concrete research proving this hypothesis is lacking and therefore genome-centric approaches to understanding the ARGs associated with size-fractionated aerosols require prioritization in apprehending ARG transmission through the inhalation–exhalation route. In addition to this, the long-range transport of these biological conjugates plays a role in their spatial distribution across geographical boundaries within and across regions. A holistic approach to mapping ARGs in the atmosphere would provide an insight into their vast potential for spatial distribution and could explain some of the underlying reasons for the seasonality of certain strains in particular regions. A better understanding of aerosol-mediated ARGs will also help in taking steps to curtail the spread of AMR by installing air purification systems in critical care units and issuing advisories to check its spread.

A systematic review of the subject was taken up using the protocols recommended by the Preferred Reporting Items for Systematic Reviews and Meta-Analyses (PRISMA). In this review, we included studies that investigated the antibiotic resistance genes in aerosols. The search terms and the inclusion and exclusion criteria were decided accordingly. The criteria used for inclusion and exclusion are graphically presented in the PRISMA flow diagram shown in Figure 1. For this systematic review, two databases were screened, PubMed and SCOPUS, using the keywords “aerosols” AND “antibiotic resistance genes” and “hospitals” AND “animal farms” AND “wastewater treatment plant”. All articles published until December 2023 were considered. The search terms built were simple to include all possible studies while maintaining the keynote of the topic. Studies were screened from the list of articles obtained from the two databases (*n* = 91 from PubMed; *n* = 125 from Scopus). After removing duplicates (*n* = 100), 116 studies were screened and review articles, systematic reviews, meta-analyses, letters to the editor, comment articles, and studies based on phenotypic resistance were excluded. A total of 57 original articles that included observational studies, comparative studies, ARGs in inhalable and respirable air fractions, long-range transport, and horizontal gene transfer were included (Appendix A). A limited number of eight studies employing shotgun metagenomics for ARG profiling, absolute quantification, and relative abundance reporting were chosen for the meta-analysis (Appendix A).

## 2. Aerosol Classification

Aerosol, in general, is defined as a suspension of solid or liquid particles in a gaseous phase. Several attempts have been made in the past to strictly define aerosols without reaching an agreement [44]. However, in a broad sense, the mixture of particulates in the air is considered as an aerosol. An aerosol can be classified based on its nature, size, and method of generation.

The nature of aerosols depends on their point source in the atmosphere such as land, seas, oceans, meteoritic flows, forest fires, and chemical and photochemical reactions. Ivelev [45] calls them soil aerosols (originating from the surfaces of steppes, deserts, and mountains), marine aerosols (solar radiation in the air over the ocean reacting with its surface layer), volcanic aerosols (volcanoes ejecting into the atmosphere), secondary aerosols (occurring due to photochemical and chemical reactions), and biogenic aerosols (organic aerosols originating as a result of volatile organic compounds).

Based on their size, aerosols are classified as fine (radius < 0.1 µm), medium (>0.1 µm to <1.0 µm), and giant (radius > 1 µm) [45]. In nature, aerosol particles exist in a wide range of sizes and diameters, ranging from 1 nm to large dust particles and droplets of water as big as 10 µm. It is difficult to predict the exact size of particles at a particular time as the existence of these particles depends on meteorological parameters. The particles of volcanic ash and dust can persist in the environment for a long time.

Aerosols may be formed naturally or through anthropogenic interventions. Naturally occurring aerosols are dust, fog, and mist. Dust is generated through the mechanical abrasion of solids or the drying of particle-laden droplets. Fog and mist are the result of condensation of liquid droplets. Human-created aerosols are smoke, perfume sprays, and irrigation mist, among others [45]. Biological aerosols are a subcategory of aerosols comprising living and non-living components of microorganisms such as fungi, viruses, bacteria, pollens, and spores [46]. Bioaerosols are typically in the size range of 0.5 to 100 µm.

Bioaerosols are known to affect living things through infectivity, allergenicity, toxicity, and pharmacological or other processes [47]. Infectious bioaerosols mostly comprise pathogenic microbes causing diseases. Exposures to such aerosols become hazardous when they carry resistance elements and make the treatment challenging. In addition to their presence, these elements can transfer from organism to organism through the mechanism of horizontal gene transfer, among living beings via the exhalation–inhalation route, and between geographical boundaries through wind carryover. In the following sections, we provide an overview of its existence and spread, highlighting the need for cutting-edge research to combat and constrain the spread of AMR.

Based on their size fractions, there is a consensus among scientists that a fraction smaller than PM_2.5_ is considered respirable, can go deep into the lungs and alveoli, and is not likely to be exhaled, while a fraction PM_10_ and above is considered inhalable and is usually is exhaled. This brings attention to the importance of size fractionation in aerosol studies and the importance of finer fractions that are likely to remain in the respiratory system for a longer time.

## 3. Indoor Aerosols Are Rich in ARBs

Humans living modern urban lifestyles spend about 89% of their time indoors [48], more so in regions where the climate is extreme (hot and cold), making indoor air quality very important. Some studies have reported that certain contaminants are higher indoors compared to the outdoors. ARGs and antibiotic-resistant bacteria (ARBs) have been identified in various indoor environments and have become occupational health hazards.

### 3.1. ARBs and ARGs in Indoor Environments of Hospitals

Indoor hospital settings are rich in ARG and ARB-laden aerosols. A high usage of antibiotics to prevent and treat all microbial infections is the most potent source of ARGs in hospices. The exhaled breath of patients undergoing treatment can be transmitted to the healthcare personnel, visitors, and other hospital staff (Figure 2a). Genes resistant to tetracycline were reported from clinics in Colorado [49]. Erythromycin and tetracycline resistance genes were also reported in the rooms and drains of a hospital in China [50]. Several ARGs resistant to multiple drugs were reported from urban hospitals in China [51,52,53,54,55,56], Hong Kong [57], Kuwait [58], and Singapore [59]. In addition to exhaled breath, occupant health, skin shedding, sneezing, apparel, personal protective equipment, and medical instruments, among others, are common sources of ARG’s deposition/aerosolization. Carpets and vinyl flooring interiors in the medical settings of Ohio State University Columbus were reported to bear rich and diverse ARGs [60]. The door handles, sinks, and floors inside a newly built hospital in Berlin accumulated ARGs over 30 weeks [61]. In many of these studies, only ARGs were mapped regardless of the microbial population hosting them. This is important in establishing inhalation dosages and risk assessment.

### 3.2. ARBs and ARGs in Animal Farm Settings

Antibiotics in livestock feed are known to result in the development of antibiotic resistance in farm animals. Sheddings from animals such as fur, feathers, saliva, and faeces are common sources of ARGs in the environment. Exhaled animal breath has been negligibly explored as a contributor and vector of ARGs in atmospheric air. Some recent research suggests that enclosures at animal farms do possess ARBs (Figure 2b). The indoor dust of a swine and poultry feeding unit possessed genes resistant to tetracycline [49]. The veterinary teaching hospital at the University of Melbourne consisted of ARGs on the inner cage surfaces, trolleys, and office floors [62]. Similarly, the interior floorings of a veterinary hospital were rich in tetracycline-, sulfonamide-, and carbapenem-resistant genes [60]. Further to these, composting plants processing animal manure possessed ARGs within the composting areas, packaging units, and offices [63]. Dust samples inside a chicken and dairy house inhabited 18 (sulfonamides—*sul1*, *sul2*; tetracycline—*tetW*, *tetC*, *tetO*, *tetH*, *tetG*; chloramphenicol—*cfr*, *cmlA*, *floR*, *fexA*; streptomycin—*aadA*; beta-lactam—*bla*_TEM_; quinolone—*qnrS*; erythromycin—*ereA*; mobile genetic elements—*Tn916*, *intl2*, *intl1*) and 16 (sulfonamides—*sul1*, *sul2*; tetracycline—*tetW*, *tetO*, *tetH*, *tetG*; chloramphenicol—*cfr*, *cmlA*, *floR*, *fexA*; streptomycin—*aadA*; beta-lactam—*bla*_TEM_; erythromycin—*ereA*; mobile genetic elements—*Tn916*, *intl2*, *intl1*) AMR elements, respectively [64]. 

### 3.3. ARBs and ARGs Inside Waste Treatment Facilities

The interiors of wastewater treatment plants have seldom been studied (Figure 2c). Indoor aerosols near the fine grid and aerated tank within a wastewater treatment plant in Qingdao, China, were reported to inhabit multidrug-resistant genes [65]. The bioaerosols proximal to secondary sedimentation tanks inside a wastewater treatment plant were positive for *bla*_CTX-M_ and *bla*_OXA_ [66]. 

### 3.4. ARBs and ARGs in Urban Atmosphere

Evidence of ARGs accumulating in the interiors of non-clinical environments has also been reported (Figure 2d). Some examples were office buildings [60,67], high school gyms [68], dormitories [69], homeless shelters [49], research laboratories [58,70], malls [52], boarding schools [52], residential houses [55], screening workshops/office areas of waste recycling sites [71], animal waste decomposition sites [Yu et al., 2021]. The types and concentrations of ARGs in these environments depend on several factors, including location, human footfall, ventilation type, and occupation.

## 4. Bioaerosols laden with ARGs in Ambient Air

ARGs are channelized from indoor to outdoor environments, and vice versa [69]. The concentrations of ARGs in ambient aerosols have been reportedly low, due to high dilution rates. However, the outdoor air in the proximity of contaminated sites such as hospitals, wastewater treatment plants, animal farms, etc. is likely to acquire the ARGs due to degassing, air circulation, and exchanges in air-conditioned complexes. Official buildings, residential townships, and urban cities near these places are vulnerable and at risk (Figure 3).

### 4.1. ARBs and ARGs Adjacent to Hospitals

Nosocomial outbreaks are well known and thus lead to the belief that hospitals and clinics are the hot spots through which ARGs are distributed to ambient air. He et al. [57] sampled the ambient air at the entrance of the outpatient department of a city hospital, the entrance of an urban community centre 0.5 Km away from the hospital, and a suburban community centre entrance (approximately 54% carriage) 10 km away from the hospital and found ARGs in all locations [57]. It was further observed that the abundance of multidrug-resistant genes reduced as they moved toward the suburban community centre [57]. ARGs were also distributed through the ventilation rooftops of the pulmonary and critical care units in China [72].

### 4.2. ARBs and ARGs across Waste Management Sites

Antibiotics administered both orally and intravenously end up in the waste disposal site through oral–faecal routes]. Improper disposal of unused or expired drugs and pharmaceuticals adds to the selective pressure imposed by these persistent pollutants. Sludge and wastewater have high concentrations of ARGs and ARBs. Not only do these antibiotics give rise to novel ARBs and ARGs but also they channel them into the atmosphere through aerosolization. Submicron aerosols above a wastewater treatment plant (WWTP) were reported to share resistomes with wastewater and sludge [41]. The total suspended particles above a WWTP consisted of multidrug-resistant and bacitracin genes [73]. The vents of three municipal solid waste transfer stations reportedly discharged 2.88 to 9.49 × 10^9^ copies m^−3^ of ARGs in the surrounding air [74].

A study from Hong Kong qualitatively and quantitatively exhibited WWTP as an important source of ARGs in urban environments. The urban air shared 57% of ARGs with aerosol above WWTP; however, the concentrations diminished as the air mass reached the coastline [75]. ARGs were demonstrated to be dispersed from swine manure biogas degassing to the atmosphere through a lab-scale dynamic emission vessel [76]. The air above the bubble aeration tank, surface agitation tanks, and site windward to a WWTP showed glycopeptide and multidrug-resistant bacterial genes, with the lowest rate ratio in upwind sites [77].

The liquid sewage released by the WWTP is also a known source of release of ARGs and ARBs in the environment. Aerosol collected both upwind and downwind of WWTP along the coast of South Carolina showed ARGs. There was a clear distinction between the ARG and ARB profiles of both sites. The downwind samples showed a high similarity in ARGs with activated sludge. The preliminary ARG dispersion model estimated an average release of ~10,700 genes per hour from the WWTP [78]. Metagenomic sequencing was able to detect ARGs in and around a full-scale WWTP [65]. ARGs were more abundant along the upwind–downwind transect. 

### 4.3. ARBs and ARGs Proximal to Animal Farms and Agricultural Sites

Yang and team [73] detected ARGs in pig and chicken farms in Zhuhai, China. The ARGs of chicken farms were more diverse than those of swine farms; is it because of extensive medication and probiotics administered to the chicks in the early life stage? It was further revealed that these ARG profiles were very similar to animal faeces and sludge. Both ARGs and ARBs emanated in the air from cattle production in a beef farm [79]. ARG (*blaSHV*, *ermF*)-impregnated bioaerosols were also determined upwind and downwind of three poultry feeding units, with the latter exhibiting higher concentrations [79]. About 10^0.5^ h^−1^ and 10^2.3^ h^−1^ of *intl1* and ARGs, respectively, were reported in the air surrounding livestock farms in China. These copies were far less than what was reported from the soil and faeces but posed a risk to human health [12].

ARGs and MGEs were found upwind and downwind of livestock farms located in the Guangdong Province of China. The number of targets were 15 and 10 in chicken, and dairy farms in the air collected 50 and 100 m upwind, respectively. Almost parallel numbers were recorded at a distance of 50 m (*n* = 15) away from chicken farms as well as 100 m (*n* = 14) downwind from the dairy farms. As expected, the targets were reduced when moving 100 m (*n* = 11) and 150 m (*n* = 11) away from the chicken house and dairy farm, respectively. In addition, *Acinetobacter* and *Staphylococcus* were shown as the two most dominant pathogens in these sites [64]. This study established through the Gaussian plume model the dispersion of ARGs and ARBs at distant locations (10 Km) to the animal farms.

### 4.4. Ambient Urban Atmosphere

A study conducted in cities in China reported that urban aerosols accommodate rich and dynamic ARGsTemperature was predicted as the key contributor to ARG variation in summer, whereas air pollution was responsible during springtime [80]. Atmospheric air collected from the public parks of California (Fresno, San Deigo, Los Angeles, Bakersfield) contained several copies of the *bla*_SHV_ (0.19–600 copies/m^3^ of air) and *sul1* (1 × 10^−2^–1 × 10^3^ copies/m^3^ of air) genes [11]. 

Open wastewater canals in cities with poor sanitation are a rich source of ARG-laden bioaerosols. A study conducted in Kanpur, India, testified that tetracyclines (*tetA*), fluoroquinolones (*qnrB*), beta-lactams (*bla*_TEM_), and class 1 integron (*intl1*) are in the air proximal to these sources. The gene copy numbers were estimated as 10^2^ and 10^3^ per cubic meter of air for ARGs and MGEs, respectively [37]. Sites 1 km away from these canals were relatively cleaner. Comparable levels of ARGs and MGEs were recorded in an almost parallel study undertaken in La Paz Bolivia [81]. Viable cells of the common sewage contaminant *Escherichia coli* were also detected in the air over the water channels. Decreasing concentrations of *bla*_TEM_, a beta-lactam-resistant gene, were observed at a distance of 150 m.

### 4.5. Other Polluted Sites

Several ARGs were reported in the outdoor premises of the Kuwait Institute of Scientific Research, Kuwait [58], located near a hospital waste discharge point on a tidal flat that is inundated only during the highest tide. Urban aerosols are also known distributors of ARGs. Aerosols near polluted riverine sites recipient to domestic, household, and emergency waste in Beijing were rich in ARGs [82]. Bacterial aerosols accumulated vancomycin and quinolone resistance genes in polluted air samples collected in Beijing City [83]. Atmospheric air upwind and downwind (250 m away) from four composting plants processing swine, cattle, and poultry manure were positive for ARGs and MGEs [63]. The numbers significantly increased in the downwind (*n* = 25) as compared to upwind (*n* = 5) direction.

## 5. Bioaerosols Accumulating ARGs in Remote/Not Impacted Locations

Apart from the contaminated sites, some studies have explored the presence of airborne ARGs in remote locations. For example, Caliz and team [84] found *sul1* (sulfonamide) *tetO* (tetracycline) and *intl1* (class 1 integron) genes in the atmospheric depositions collected from the free tropospheric layers of remote mountains located in Pyrenees, Spain. The tetracycline-resistant genes were traced back to African dust outbreaks, suggesting the long-range transport of ARGs. This study highlights the importance of assessing the air trajectories as a route to intercontinental transfer. The bacterial communities correlated with the agricultural soil. Two tetracycline-resistant genes were tested in the alpine forests of Colorado, and none were detected. Interestingly, the site was positive for the class 1 integron gene [49]. ARG was also identified in the dust accumulated on air-conditioning filter units in three village houses in China. The copy numbers in rural houses (3.29 × 10^−3^ copies/16S rRNA) were relatively higher than their city (3.64 × 10^−6^ copies/16S rRNA) counterparts [55]. 

## 6. ARGs in Inhalable Fraction of Air

Particulate matter (PM) in a size fraction < 2.5 µm, which is usually reported as PM_2.5_, is an inhalable fraction of aerosol, and the presence of ARGs therein poses considerable risk compared to larger aerosol size fractions. The presence of ARGs in PM_2.5_ is likely to increase the chances of ARG dissemination due to inhalation and leads to reduced efficacy towards the treatment of respiratory infections. Tetracycline resistance genes and class I integrons were found in the indoor and outdoor PM_10_ aerosols of a clinic, homeless shelter house, animal feeding farms, livestock agriculture units, and the alpine forests of Colorado [49]. About eight dominant ARGs were identified in the PM_2.5_ collected over the WWTP [75]. Although the group reported a low resistome risk score associated with PM_2.5_, they highlighted that pathogens harbouring ARGs in atmospheric aerosols pose a significant risk to human health.

Similarly, ARGs were detected in the PM_2.5_ aerosols collected from industrial, urban, and rural sites in Beijing, China [82]. The authors of this study demonstrated aerosol as a plausible way to spread ARGs among the human population. Aerosols with PM_2.5_ and PM_10_ contributed significantly towards ARG accumulation in urban cities of China in spring [80]. ARGs were reportedly consistent in PM_2.5_ and PM_10_ and posed a great risk to human health [57]. Air particulate matter with PM_2.5_ and PM_10_ significantly increased the conjugative plasmid transfer rate mediating the spread of ARGs by 110% and 30%, respectively [85]. Hospital PM_2.5_ reportedly harboured ARGs with a resistome risk index of 21.17, significantly higher than the ambient urban risk ratio [72].

Workers in the medicare industry, and animal husbandry are at high occupational risk, owing to the inhalation of ARG-laden bacteria and fungi. The abundant distribution of ARGs among aerosol and human nasopharynx was very similar in a chicken farm environment [86]. Likewise, the interiors of crowded places are hypothesized as hot spots of ARGs. Antibiotic resistome was found prevailing within PM_1.0_ in the indoor-outdoor aerosols as well as the upwind–downwind transect of a wastewater treatment plant in Qingdao, China. These ARGs were resistant to multiple drugs and associated with plasmids and transposons mediating its spread [65]. 

## 7. Microbial Hosts

The global action plan for AMR was adopted by the World Health Organization (WHO) in 2016, and the first bacterial priority pathogen list (BPPL) was released. This encompassed pathogens such as *Acinetobacter baumanii* (carbapenem-resistant), *Enterococcus faecium* (vancomycin-resistant), and *Streptococcus pneumonia* (penicillin-resistant) [87]. Antibiotic resistance is continually evolving, and many more pathogens are being added to the list, including *Escherichia coli*, *Staphylococcus aureus* (methicillin-resistant (MRSA)), *Klebsiella pneumonia*, and *Pseudomonas aeruginosa*. The new BPPL is yet to be released, and all the above bacteria are collectively listed as ESKAPE pathogens [87]. In 2019, these pathogens were responsible for more than 2 million deaths across the world [88]. It is, therefore, of paramount importance to isolate and characterize these pathogens for ARGs, mobile genetic elements, and gene cassettes, the causal elements behind their evolution.

The ESKAPE pathogens and MDR bacteria are an interconnected global threat as they can spread between organisms, environments, and ecosystems [89]. High rates of resistance against the common antibiotics used to treat urinary tract infections, sepsis, respiratory diseases, diarrhoea, etc. have been observed globally within hospital environments. For instance, the rate of resistance to ciprofloxacin, frequently used to treat urinary tract infections, ranged from 8.4% to 92.9% for *Escherichia coli* and from 4.1% to 79.4% for *Klebsiella pneumoniae*, resulting in 23% mortalities [90]. Some specific examples of each strain within hospital settings have been discussed in the following text.

### 7.1. Enterococcus faecium

*Enterococcus faecium* is an opportunistic pathogen commensal to human and animal gut. Since the increased usage of antibiotics over the past decades, it has developed acquired resistance and has been classified as a common hospital dweller [91]. Several vancomycin-resistant clones (VREfe) are known to be in circulation within hospital settings [92]. A five-year retrospective study identified about 18.7% of MDR bacteria as VREFe in an oncology unit in Mexico [90].

### 7.2. Staphylococcus aureus

*Staphylococcus aureus* is a notorious hospital occupant and well known for resistance against methicillin (MRSA), a beta-lactam antibiotic [93]. A high rate of MRSA (94.05%) was reported in a tertiary care hospital in Nepal [94]. Approximately 90% of MRSA was found in the oncology unit of a teaching hospital in Mexico [90]. MRSA has also been isolated from the surfaces of the internal medicine unit, surgery, and intensive care unit of an equine teaching hospital at Ohio State University [95] and a 1200-bed teaching hospital in London [96]. It is a common inhabitant of the nasal mucosa, and hospital-acquired carriage by healthcare occupants has been well documented [97,98,99,100,101,102].

### 7.3. Klebsiella pneumonia

*Klebsiella pneumonia* is another multidrug-resistant pathogen responsible for morbidity and mortality *in lieu* of limited treatment options in healthcare-associated infections [103]. Nosocomial outbreaks of *Klebsiella* have been reported in China [104] and Portugal [105]. The evolution of this genus as an MDR strain has been reviewed extensively by Venezia et al. [103].

### 7.4. Acientobacter baumanii

*Acinetobacter baumanii* poses a great challenge for clinicians as it is very commonly found in intensive care units and medical devices [106]. Eventually, *A. baumanii* developed resistance against a broad spectrum of antimicrobials. About 6% of the bacterial strains were identified as MDR *A. baumanii* in a Mexican hospital [90].

### 7.5. Pseudomonas aeruginosa

*Pseudomonas aeruginosa* is a Gram-negative bacterial pathogen known to cause severe nosocomial infections [107,108,109,110]. The strains easily acquire resistance through mutations and develop genes encoding beta-lactams and aminoglycoside-modifying enzymes [107]. The overall economic cost to treat multiple drug-resistant *Pseudomonas aeruginosa* (MDRPA) was increased by 70% in a hospital in Barcelona [108].

### 7.6. Enterobacter sps.

*Enterobacter* is the most complex ESKAPE pathogen, consisting of a group of 22 species exhibiting resistance against multiple drugs [111]. *Enterobacter aerogenes* and *E. cloacae* jointly contributed to several hospital outbreaks [112,113]. Fourteen isolates of Enterobacteriaceae origin (*Enterobacter cloacae*, *Escherichia coli*, and *Citrobacter freundii*) were found in WWTPs [66].

## 8. Horizontal Gene Transfer

The phenomenon of horizontal gene transfer (HGT) has been commonly reported in aquatic and terrestrial environments [26,114]. The inherent capability of microbes to form biofilms on the available substrates is key to the high rate of HGT in these environments. Biofilm formation in the air is relatively rare; however, bioaerosols are a conjugate of bacterial, fungal, and viral communities and their metabolites. The exchange of genetic material is quite plausible within these microstructures. The plasmids, transposons, integrons and mobile genetic elements (MGE) play major roles in the carriage of ARGs to non-resistant bacteria (Figure 4). Multi-resistant plasmids were prevalent in hospital particulate matter in China [85], with a significant increase in conjugation rate by 110% by PM_2.5_. Mobile genetic elements (*intI1*, *tnpA-02*, *tnpA-04*) were also observed in areas adjoining the Yangtze River and the Pearl River delta of China [82]. Class 1 integron *intl1* and mobile genetic elements were associated with *sul1* and *aadA* ARGs in air particulate matter over a WWTP [75]. Three ARGs were found on the plasmid DNA of bacteria isolated from indoor office aerosols in Poland [67]. MGEs were associated with sulfonamide, tetracycline, and bet-lactamase genes in dust collected from dense urban public places [52]. Multiple antibiotic-resistant genes inhabiting plasmid sequences (*n* = 5910) and bacteriophages (*n* = 1693) were assembled from 179 sites over 1.5 years in a tertiary care hospital [59] (Table 1). Similar observations were recorded from a WWTP in China [65].

## 9. Aerosol-Mediated AMR Implications in Kuwait

The aerosol-mediated spread of AMR calls for attention in a country like Kuwait. ARGs are hosted within the microbial communities, and our group has previously demonstrated aerosols as the vectors for bacteria, fungi, and viruses in hospitals and other indoor environments [31,32,33,34,35,42,115]. Not only have these several opportunistic pathogens been identified in the ambient air [31,42,116,117,118,119], but also we have explored the nasal microbiomes of healthcare laboratory staff and found several bacterial genera common to the aerosols [97,120]. The signatures of ARGs in indoor and outdoor aerosols within urban and hospital settings have also been recorded [58]. ARBs and ARGs were channelized in marine sediments through emergency waste discharges [14,15,24,25,121,122,123,124]. It is quite plausible that they are aerosolized in the nearby environment. Integrated monitoring is therefore of utmost importance in Kuwait’s atmosphere.

## 10. Meta-Analysis of ARG Abundances

A meta-analysis was performed on eight selective studies employing the shot-gun metagenomics method to capture the ARG counts and their relative abundances. qPCR-based estimations were excluded from the meta-analysis as it is based on targeted genes and are inexpressive of the overall abundances [125]. The meta-analysis supported our observations of the ARG densities being higher near the animal farms, wastewater treatment plants, and animal farms (Figure 5a). The relative abundances followed the same trend (Figure 5b).

Microbiological quality index of indoor environment were reportedly higher as compared to outdoors [126,127,128]. Seasonality [129], weather conditions [130,131], and environmental parameters [132] plays an important role in defining the resistomes. The fate, distribution and behaviour of ARGs is activity, and location driven [133,134,135,136,137,138,139,140,141,142].

## 11. Conclusions

Multidrug resistance is an under-appreciated public health threat around the globe. Limited studies have reported ARGs in aerosols, and even more scarce are the ones that have source apportionment. Three infectious syndromes dominated the global burdens of AMR deaths, and the highest were lower respiratory and thorax infections [88]. This suggests exhalation and inhalation as point sources of MDR microbes in the atmosphere, and the occupants of such environments are therefore at a health risk. There is also a high likelihood that these pathogenic microbes transmit the ARGs to other species in close association via the mechanism of horizontal gene transfer. A repertoire of pathogenic and non-pathogenic multiple drug-resistant microbes is speculated to prevail in the indoor atmosphere, and transmission to ambient locations is plausible.

The issue of AMR is garnering international attention on account of the health hazard imposed. The aerosol-mediated spread of AMR is a limitedly researched area under the “One Health” umbrella. Although initial steps have been paved, deeper investigations at the global and regional levels are required. Whether inhalation will cause acute health threats needs further investigation, but continual exposure cannot be ignored, especially in the case of immunocompromised individuals. Regular monitoring and surveillance are thus recommended. The wind air trajectories should be looked at, and transboundary transmission should be investigated. The microbiomes of the occupants, workers, and their families, as well as their residences need to be checked. Moreover, antibiotic stewardship programs should be launched to educate healthcare practitioners and healthcare seekers.

## Figures and Tables

**Figure 1 ijerph-21-00983-f001:**
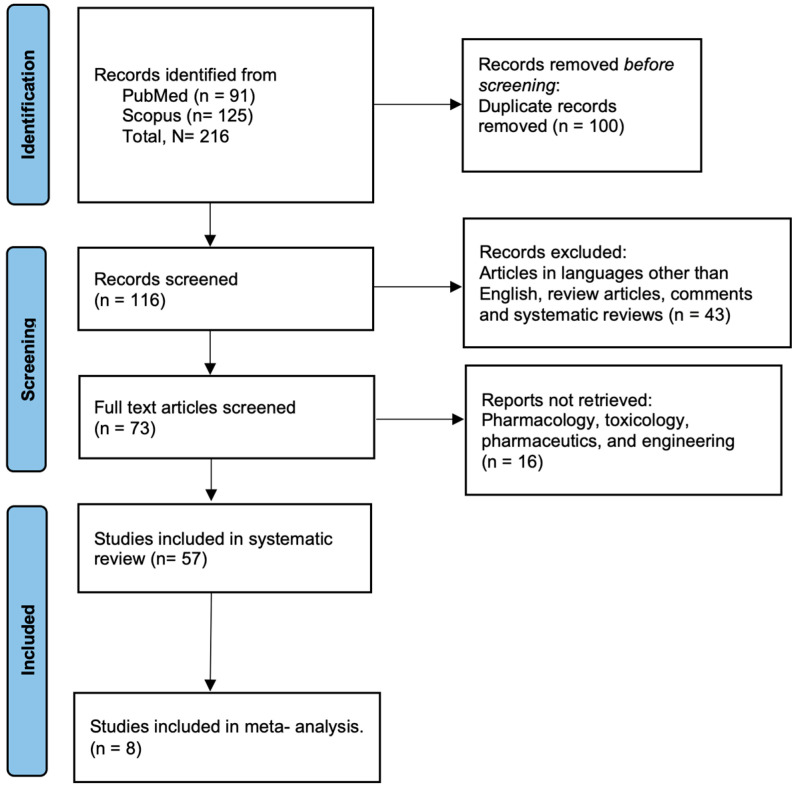
PRISMA flow diagram depicting the search criteria.

**Figure 2 ijerph-21-00983-f002:**
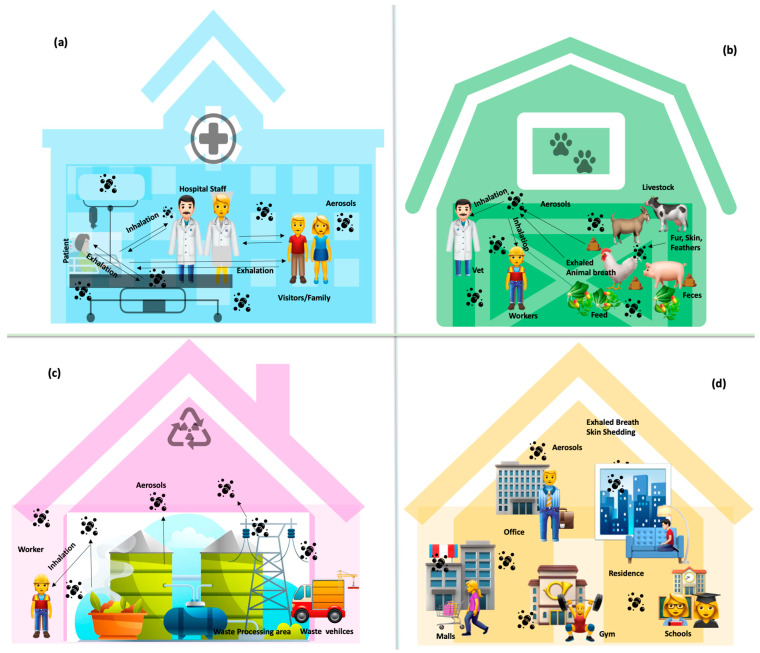
Possible aerosol-mediated spread of ARGs in indoor environments: (**a**) hospitals, (**b**) animal farms, (**c**) waste management sites, and (**d**) urban atmosphere.

**Figure 3 ijerph-21-00983-f003:**
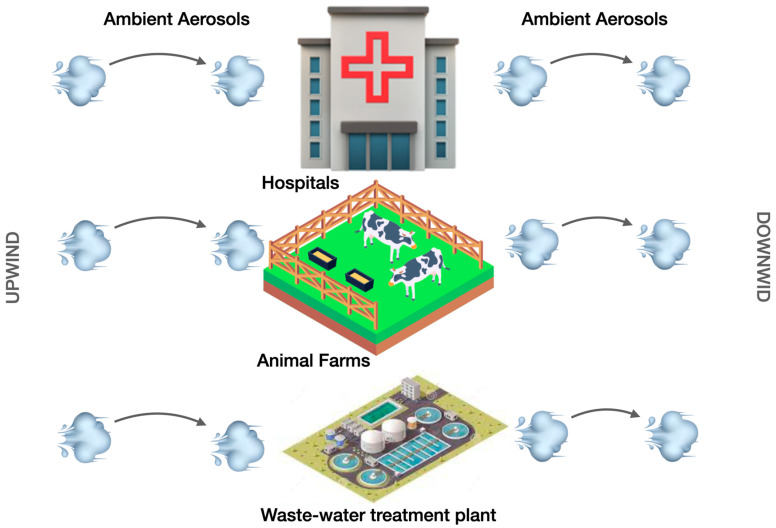
Long-range transport of ARGs from the upwind to downwind directions.

**Figure 4 ijerph-21-00983-f004:**
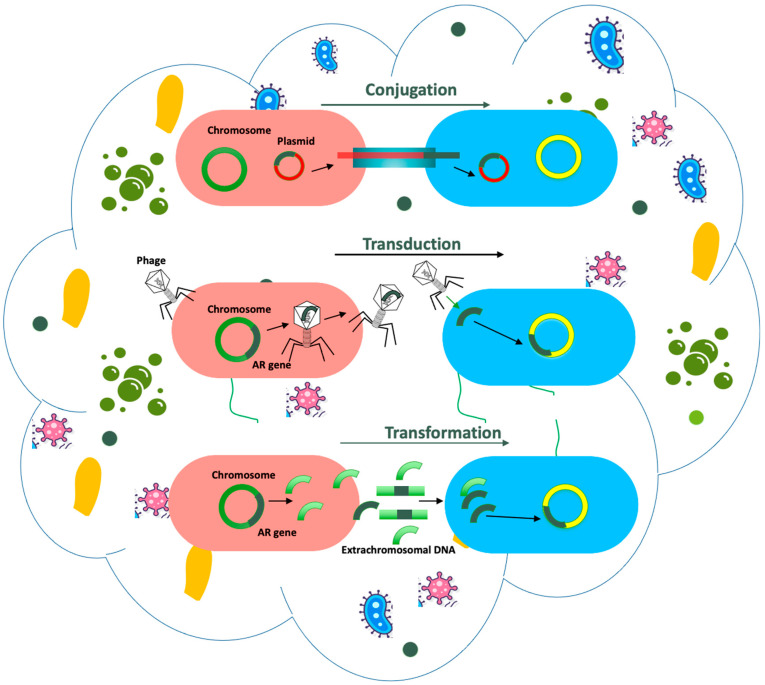
Schematics of horizontal gene transfer within a bioaerosol.

**Figure 5 ijerph-21-00983-f005:**
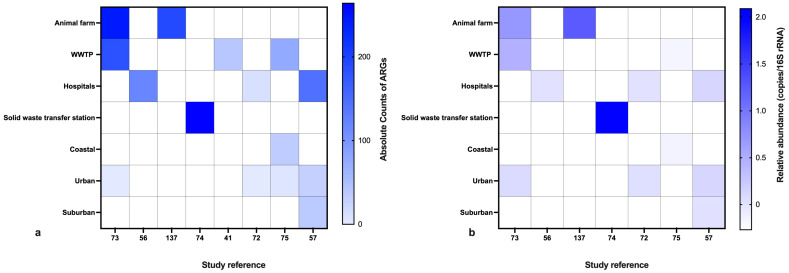
Meta-analysis of ARG abundances: (**a**) absolute counts; (**b**) relative abundances.

**Table 1 ijerph-21-00983-t001:** Antibiotic resistance gene elements mediating HGT in aerosols.

HGT Elements	Sample	Reference
26 horizontal transfer elements (*intl1*, *Tn3*, *TnAs1*, *TnAs3*)	Hospital aerosol	[85]
*intI1*, *tnpA-02*, *tnpA-04*	Riverine atmosphere	[82]
Mobile genetic elements, *intl1*	WWTP	[75]
*ermA*, *aacA-aphD*, *mecA*, *tetK* on plasmid DNA	Office rooms	[67]
Mobile genetic elements	Office dust, malls, hospitals, schools, parks	[52]
Plasmids and phages	Tertiary care hospital	[59]
Plasmids and transposases	Full-scale waster water treatment plant	[65]

HGT—horizontal gene transfer.

## Data Availability

No research data was generated in this manuscript.

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
