# Peer review of "Aerosol-Mediated Spread of Antibiotic Resistance Genes: Biomonitoring Indoor and Outdoor Environments"

_ijerph, 2024, doi:10.3390/ijerph21080983_

Round 1

Reviewer 1 Report

Comments and Suggestions for Authors

Dear Authors,

Your review entitled "Aerosol Mediated Spread of Antibiotic Resistance Genes: Biomonitoring the Indoor and Outdoor Environments" has been reviewed,

This manuscript deserves attention since it highlights an important topic related to occupational health and public health. In fact AMR is one of the leading fields of interest from medical and research points of view. Tackling different way of transfer of antimicrobial resistance genes is very important and to know were and how these genes are transferred is very important in fighting AMR. Therefore this review is very important for researchers and physicians at the same level.

The paper is well written in English, well presented, the figures are very demonstrative and attractive for readers.

Here is the list of my minor comments:

01- In the Abstract, line 30, Authors are invited to replace "the antimicrobial resistance genes (ARGs)" by "ARGs".

02- In line 36, authors are invited to put the subtitle with the graph on the same page.

03- In the Introduction, line 42, when authors talked about the use excessive use of antibiotics, they are invited to specify that it is more related to the excessive use of broad spectrum antibiotics.

04- In the Introduction, lines 71-13, when authors talked about the importance to fight AMR they are invited to use the following two articles as references for this point:

-- https://www.sciencedirect.com/science/article/abs/pii/S0882401023002541

-- https://www.ncbi.nlm.nih.gov/pmc/articles/PMC9047147/

05- In lines 93-95, "Exhaled... (Fig. 2b)" Authors are invited to add the following article as reference for this point:

-- https://jidc.org/index.php/journal/article/view/32032018

06- Concerning the Figure 2, I suggest to move it after the line 130.

07- Concerning the sentence between lines 263-265, Authors are invited to put references for this idea, I suggest the following articles as references:

-- https://www.mdpi.com/2076-3271/8/3/32-- 

-- https://www.ncbi.nlm.nih.gov/pmc/articles/PMC8575383/

08- When authors talked about Staphylococcus in the line 287, authors are invited to add the following paper as reference for this idea: 

-- https://www.mdpi.com/2079-7737/11/10/1525

09- When authors talked about Klebsiella in the line 297, authors are invited to add the following paper as reference for this idea: 

-- https://www.mdpi.com/2079-7737/13/2/78

10- When authors talked about Klebsiella in the line 309, authors are invited to add the following paper as reference for this idea: 

-- https://www.mdpi.com/2079-7737/11/12/1711

I Just have one Major Remark:

Why you did not apply a systematic review and Meta-Analysis design? It would be of a better scientific impact. 

BR,

Reviewer 2 Report

Comments and Suggestions for Authors

The article was studied. The medical aspect of the issue is not in my expertise, but in terms of aerosol, the following points can be reviewed and mentioned:

1. There are different types of aerosols, and it is better to classify them first.

2. Each type of aerosol may have different roles in the article's subject, which need to be investigated.

3. It is better to examine the article with real examples and the effect of geographical factors on them.

Comments on the Quality of English Language

-

Author Response

See the responses in the attached file.

Round 2

Reviewer 1 Report

Comments and Suggestions for Authors

Dear Authors,

I would like to thank you the Reviewer for the modifications they did, the article is way better in its present form,

Otherwise, I am still with the point that doing a meta-analysis is better.

BR,

Author Response

We are thankful for your review and appreciation that the quality and readability of the manuscript have improved. The review was done considering the PRISMA guidelines however we were of the view that since the data is summarized in the table we need not elaborate on meta-analyses. On your suggestion again during this review, we have decided to comply with the recommendation and have made the required adjustments to the revised manuscript. We hope you will find it acceptable now.

Reviewer 2 Report

Comments and Suggestions for Authors

All the points in the article have been corrected and the article is acceptable in my opinion.

Comments on the Quality of English Language

-

Author Response

Thanks for accepting the manuscript in its current form